# Meteorological Satellite Observations Reveal Diurnal Exceedance of Water Quality Guideline Thresholds in the Coastal Great Barrier Reef

Larissa Patricio-Valerio [1,2,*], Thomas Schroeder [2], Michelle J. Devlin [3], Yi Qin [4] and Scott Smithers [1]

1   College of Science and Engineering, James Cook University, Townsville, QLD 4811, Australia
2   Commonwealth Scientific and Industrial Research Organisation, Environment, GPO Box 2583, Brisbane, QLD 4001, Australia
3   Centre for Environment Fisheries and Aquaculture Science, Parkfield Road, Lowestoft, Suffolk NR33 0HT, UK
4   Commonwealth Scientific and Industrial Research Organisation, Environment, GPO Box 1700, Canberra, ACT 2601, Australia
*   Correspondence: larissa.patriciovalerio@my.jcu.edu.au

**Abstract:** The Great Barrier Reef (GBR) is a marine protected area subject to natural and anthropogenic disturbances. Water quality is critical for the health and protecting resilience of GBR coral ecosystems against the synergistic and cumulative pressures of tropical cyclones, marine heat waves, and outbreaks of crown-of-thorns starfish. The concentration of Total Suspended Solids (TSS) is a key water quality parameter measured at multiple spatio-temporal scales from in situ probes to satellite observations. High TSS concentrations can adversely impact coral and seagrasses on the inshore GBR. We present diurnal TSS derived from Himawari-8 Geostationary satellite observations at 10 min frequency and demonstrate its applicability for improved monitoring of GBR water quality. Diurnal TSS obtained from Himawari-8 observations were compared to TSS computed from in situ bio-optical measurements at the Lucinda Jetty Coastal Observatory (LJCO). The coastal waters at LJCO experience diurnal variability of TSS (~7 mg L$^{-1}$), where magnitude peaks followed the slack tides, and the largest diurnal changes were associated with freshwater discharge residuals from the wet season. Exceedance maps revealed that TSS is above guideline thresholds in the open coastal and mid-shelf waters for ~60% of the valid monthly observations, including during dry season months.

**Keywords:** Himawari-8; Total Suspended Solids; bio-optics; backscattering; diurnal ocean colour; water quality; Great Barrier Reef

## 1. Introduction

The Great Barrier Reef (GBR) of Australia is an iconic, biodiverse, and complex ecosystem. With more than 3000 coral reefs distributed along the Queensland coastline (~2300 km), the GBR is the only living structure visible from space. Due to its intrinsic ecological value, the GBR Marine Park was listed as a World Heritage Site to be managed and protected for future generations [1,2]. However, despite this protection, the GBR has been negatively affected by the direct and indirect impacts of natural and anthropogenic pressures such as climate change [3,4] and declining water quality [5–7]. Water quality decline is commonly associated with modified catchment land-use and increases in sediment and contaminant loads in wet season discharges to the GBR lagoon [8–13].

The GBR and its catchments are subject to tropical and subtropical monsoonal climates with distinct dry and wet seasons. 60 to 80% of the annual rainfall (>1000 mm) occurs between November to April, often associated with the passage of tropical cyclones and monsoonal through [14], coupled with El Niño southern oscillation (ENSO) variability [15]. Thirty-five major rivers drain into the GBR lagoon, comprising the largest external source of 'new' nutrients to the system [16]. The mainland catchments have a combined area of

423,070 km², about 25% of the State of Queensland (Figure 1—left panel), with topographic features including a range of hills and low mountains (mostly within 1000 m altitude) distributed along the narrow coastal plains [14]. The largest catchments to the GBR, such as the Burdekin, Fitzroy, and Herbert (Figure 1—right panel), however, drain extensive areas through topographical gaps in this coastal range. These floodplains and low mountain ranges are particularly suited for sugar cane cropping and cattle grazing, which accounts for ~70% of the land use in the GBR catchments [17].

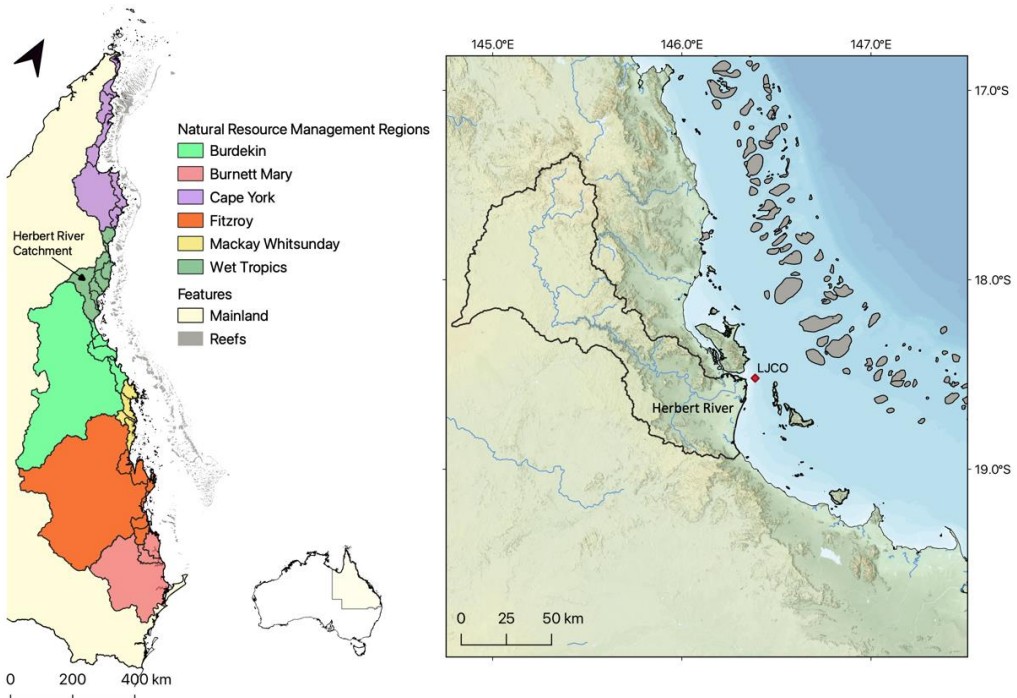

**Figure 1.** Mainland Natural Resource Management (NRM) regions and catchments draining to the Great Barrier Reef (**left** panel), Australia. Detailed relief map of the Herbert River catchment area, including the Herbert River watercourse and the Lucinda Jetty Coastal Observatory (LJCO) location (**right** panel).

Historical large-scale land clearing and subsequent erosion of bare grazing lands greatly amplified sediment exports from the GBR catchments to the adjacent coastal waters [17–19]. In fact, terrestrial sediment and nutrient concentrations annually entering the GBR have increased by more than 5-fold since European settlement [14,20–22]. Acute and chronic exposure of coral reefs to excess nutrients and fine suspended sediments is linked to a rise in coral mortality rates, as well as compromised resilience and recovery following disturbance events [23–30]. Likewise, declining water quality negatively impacts seagrass abundance [31–33] and may be a key factor in the outbreaks of coral-eating crown-of-thorns starfish [34–36]. High-frequency (minute to hourly) coastal processes, such as tides, winds, episodic floods, and algal blooms, also occur on the GBR [37,38]. These are known to regulate the fluctuations of key water quality parameters, including phytoplankton biomass and particle composition [39–42]. In this context, a comprehensive and continuous monitoring system is imperative to effectively assess coastal water quality, compliance with environmental regulations, and the success of ongoing management interventions in the GBR [43].

Remote sensing observations, biogeochemical modelling, and in situ water quality data-sampling have been extensively employed to monitor the GBR from daily to inter-annual temporal scales [9,42,44–51]. For instance, contemporary Low-Earth Orbit (LEO) satellites, such as the Sentinel-2 Multi-Spectral Instrument (MSI), Sentinel-3 Ocean-Land Colour Instrument (OLCI), Aqua's Moderate Resolution Imaging Spectroradiometer

(MODIS), the Suomi National Polar-Orbiting Partnership (SNPP) and the Joint Polar Satellite System (JPPS) Visible Infrared Imaging Radiometer Suite (VIIRS) sensors provide daily ocean colour observations to estimate water quality and to track and monitor freshwater discharge into the GBR [44,45,52]. However, because of their characteristic orbital scheduling and swath, the LEO ocean colour sensors scan the same geographic area within one or two days at best. Furthermore, the time-lag between two consecutive and identical orbits (i.e., revisit periodicity) is usually one or two weeks. Coastal areas are consequently observed during different tidal stages by the LEO satellites, which are too coarse to effectively monitor dynamic coastal processes with diurnal or sub-diurnal variability [53,54].

The contrasting optical features between the estuarine and coastal waters of the GBR are illustrated in Figure 2 at the Lucinda Jetty Coastal Observatory (LJCO, 18.5°S, 146.4°E). The images were acquired 5 days apart during the neap (Figure 2a) and spring tides (Figure 2b) with ranges of 1.2 m and 2.4 m, respectively. The colour variations of these events illustrate the dynamic complexity of the coastal GBR and the difficult task of timely collecting reliable estimates of materials exported from the catchments [55]. Hence, there is a real need to systematically monitor the state of water quality on a diurnal scale for a complete understanding of baseline conditions and trends in the GBR [56–58].

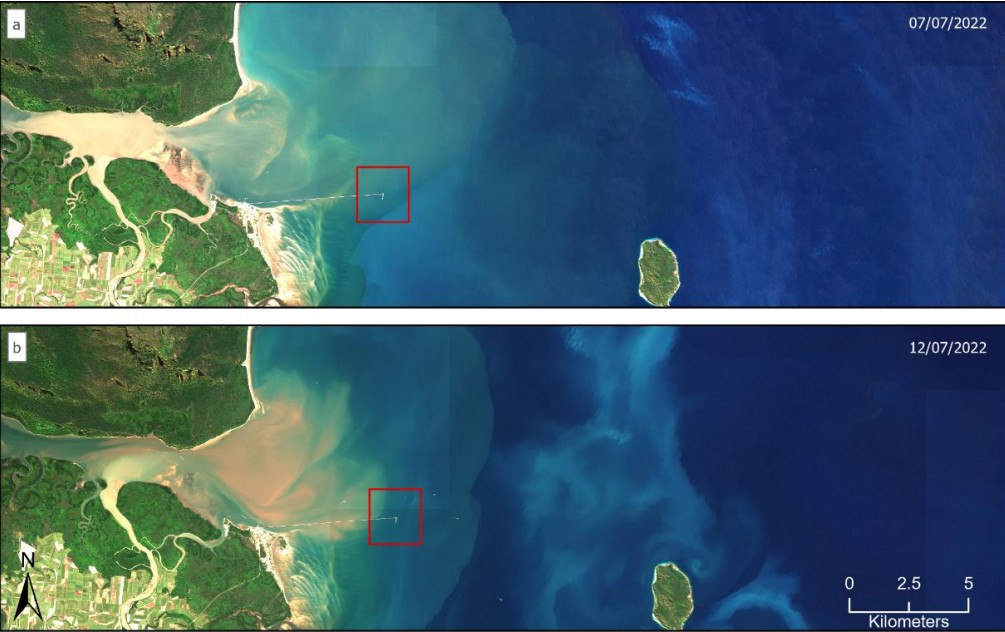

**Figure 2.** True colour composites of Sentinel-2A (**a**) and 2B (**b**) Multi-Spectral Instrument observations at 10 m resolution were acquired on the 7 and 12 July 2022 (**a** and **b**, respectively) at around 10:30 am local time. The location of the Lucinda Jetty Coastal Observatory (LJCO) site is indicated within the red rectangle. Image courtesy of the European Space Agency.

The Advanced Himawari-8 Imager (AHI) is a Geostationary-orbit (GEO) meteorological satellite sensor scanning full disk images of Australia and East Asia and offers the possibility to investigate GBR-wide fluctuations of coastal water quality from 10-min to hourly temporal resolution and beyond. At least 48 images per day can be obtained from Himawari-8, which under cloud-free conditions, may result in an up to fifty-fold increase in observations compared to contemporary LEO ocean colour sensors. In addition, Himawari-8 AHI offers a moderate spatial resolution of 0.5 to 1 km, compatible with contemporary (Sentinel-3 OLCI at 0.3 km) and legacy (MODIS/Aqua 0.25, 0.5, and 1 km) ocean colour missions, which is adequate to monitor large coastal areas, such as the GBR. High spatial resolution (10 m) observations from contemporary multispectral sensors, such as the Sentinel-2 MSI, illustrate fine-scale detail ocean colour features (Figure 2), but at a weekly frequency. Despite its limitations as a meteorological sensor (i.e., signal-to-noise

ratios and spectral characteristics, described in detail in Patricio-Valerio et al. [57] and Patricio-Valerio [59]), Himawari-8's high temporal frequency provides significant advances for quantifying and monitoring coastal water quality dynamics in the GBR [38,48,60]. Diurnal optical properties from ocean colour remote sensing may be assimilated for refined parameterization and validation of biogeochemical models and deepen our understanding of water quality impacts on ecosystems [47,61,62].

Quantitative and qualitative information about coastal water quality and plume evolution in the GBR is now available at 1 km$^2$ spatial resolution from the 10-min Himawari-8 AHI observations and the TSS algorithm developed and validated in Patricio-Valerio et al. [57]. Here we used the machine learning algorithm developed by Patricio-Valerio [59] and Patricio-Valerio et al. [57] to derive TSS from Himawari-8 AHI observations for an integrated assessment of diurnal water quality fluctuations in the coastal GBR. Continuous TSS concentrations computed from bio-optical data collected at the LJCO allowed the comparison with Himawari-8-derived TSS for an integrated assessment of their temporal coherency and to detect drivers of diurnal variability. Maps of maximum diurnal changes and the frequency with which the water quality guidelines were exceeded reveal temporal and spatial patterns of TSS variability in the coastal GBR. We provide recommendations for improved monitoring and for potentially revising the GBR water quality guidelines and progress metrics.

## 2. Methods

### 2.1. The Lucinda Jetty Coastal Observatory

The Lucinda Jetty Coastal Observatory (http://lucinda.it.csiro.au/, accessed on 23 April 2023), hereafter LJCO, was established to support the validation of satellite ocean colour radiometry and to help advance the understanding and link between the radiometry and the inherent optical properties to support the study of biogeochemical processes and modelling. The site, funded under the Integrated Marine Observing System (IMOS), is currently the only fixed-platform in Australia that does sustained observations of this nature.

The LJCO is located at the end of a 5.8 km long jetty, 12 km from the mouth of the Herbert River. The Herbert River basin is the fourth largest catchment of the central GBR (~10,000 km$^2$) and the largest river system of the Queensland Wet Tropics region, with a mean discharge of ~500,000 megalitres a day (Figure 3) during extreme wet years [63]. The Herbert catchment experiences variable rainfall distribution, with the upper western portion being the driest (~700 mm/year) and the eastern coastal plains the wettest (~3000 mm/year). At the peak of the wet season, between December and March, the lower flood plains may be subject to ~600 mm of rainfall within just a few days [64]. Marine water quality fluctuations at the LJCO are strongly associated with the hydrodynamics of the Hinchinbrook tidal channel, which includes a large mangrove area and the Herbert River mouth [42,65]. Semi-diurnal tides, wind-driven circulation, and interannual and seasonal fluctuations of freshwater discharge largely affect the seaward advection and evolution of the Herbert River plume (plume, hereafter) [50,65]. Plume waters presenting TSS of ~30 mg L$^{-1}$ and Coloured Dissolved Organic Matter (CDOM) absorption (443 nm) of up to 6 m$^{-1}$ may reach the LJCO location and the mid-shelf reefs, particularly during wet season floods (November to April) [42,55,66]. In the absence of floods, particularly during the dry season (May to October), TSS values may fluctuate between ~2 and 23 mg L$^{-1}$, depending on the tidal phase (ebb and flood), lunar cycle (spring or neap) and local winds [42,67]. The plume may rapidly change its dispersal direction (North to South) within a day in response to changing winds [55,68]. During strong south-easterly winds, the plume is generally restricted near-shore and moves northward [55,69], whereas, during northeast winds, plume waters are pushed southward and occasionally mix with waters advected from the Burdekin [70]. Under calm wind conditions, however, tidal currents produce a cross-shelf seaward advection of the plume. Turbidity plumes in the GBR may travel further than 100 km from the stream mouth, but derived material, such as particulate and dissolved substances, may be detected nearly 200 km away from the terrigenous source [10,55,69].

### 2.2. Water Quality Guideline Threshold Values for the Great Barrier Reef Marine Park

The GBR inshore water quality is assessed on an annual basis [71] to measure progress toward the Reef 2050 Water Quality Improvement Plan [43]. Guideline threshold values were developed by De'ath and Fabricius [72] as a framework to interpret the water-quality measurements obtained within the Marine Monitoring Program [73]. Three main cross-shelf GBR Marine Water Bodies are illustrated in Figure 3, and their respective guideline threshold annual and seasonal TSS values are compiled in Table 1. The open coastal and mid-shelf waters encompassed identical guideline values for TSS and were therefore considered as one water body here. Seasonal adjustments to the annual values are provided as wet and dry season values, except for the enclosed coastal boundary.

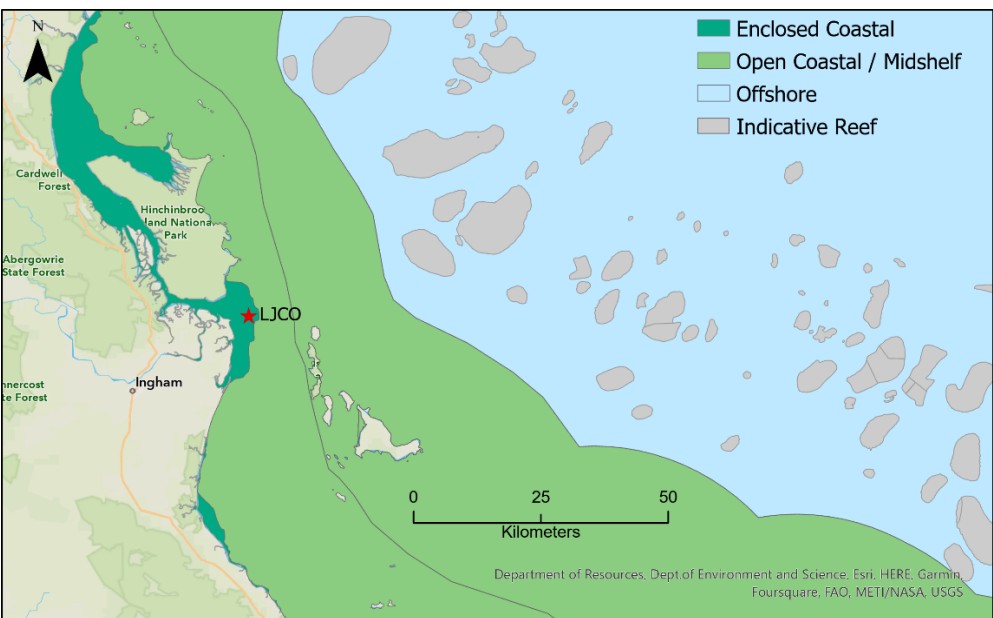

**Figure 3.** Marine Water Bodies delineation for the Enclosed Coastal, Open Coastal, Midshelf, and Offshore waters of the Great Barrier Reef Marine Park. The location of the LJCO is indicated. Data sourced from the Great Barrier Reef Marine Park Authority Database.

**Table 1.** Annually and seasonally adjusted (±20%) TSS guideline threshold values (in mg L$^{-1}$) for the GBR Marine Water Bodies, sourced from the Great Barrier Reef Marine Park Authority [73]. Enclosed Coastal thresholds depend on the catchment definition, and seasonal values are not available (N/A).

|  | Enclosed Coastal | Open Coastal/Midshelf | Offshore |
|---|---|---|---|
| Annual mean | 5 and 15 | 2.0 | 0.7 |
| Wet Season | N/A | 1.6 | 0.8 |
| Dry Season | N/A | 2.4 | 0.6 |

The enclosed coastal waters of the GBR are defined as estuarine and shallow coastal areas subject to frequent exchange and mixing between fresh and marine waters due to tidal variations [74]. The LJCO facility is located within the enclosed coastal waters of the GBR lagoon, in the Wet Tropics catchments (including the lower Herbert), and in Central Coast (from Port Douglas to Whitsundays). In these areas, annual average TSS guideline thresholds range from 5 mg L$^{-1}$ in the Wet Tropics to 15 mg L$^{-1}$ for the Central Coast, where seasonally dry tropical conditions are typical. The guidelines suggest an annual average TSS of around 2 mg L$^{-1}$ for open coastal and midshelf waters of the GBR lagoon and lower than 0.7 mg L$^{-1}$ in the mid-outer shelf zone where most reefs are located.

### 2.3. Modelling the Total Suspended Solids from Particulate Backscattering Data

Continuous in situ measurements of subsurface bio-optical properties, such as backscattering and turbidity, quantitatively describe diurnal water quality changes at the LJCO [42]. In addition, fortnightly in situ, gravimetric TSS measurements have been acquired at this site for water quality monitoring and ocean colour validation investigations [52], providing additional TSS data for modelling its relationship to the bio-optical quantities and for deriving continuous TSS at LJCO [42].

The particulate backscattering coefficient ($b_{bp}$) is a surrogate for particulate matter or TSS that can be derived from empirical formulations at this site [75,76], showing reasonable correlations despite the limitations of both measurement methods [42,77]. Soja-Woźniak et al. [42] derived a highly correlated relationship between the daily averages of $b_{bp}$ (595 nm) and gravimetric TSS from 34 pairs of measurements collected between 2014 and 2016 at the LJCO. This relationship was reviewed here, utilising a larger dataset (2014 to 2019) of nearly-concurrent in situ measurements of $b_{bp}$ (595 nm) and gravimetric TSS. This relationship was then used to compute TSS from the daily and continuous $b_{bp}$ measurements at 4 Hz, averaged to 10 min intervals, and compared against concurrent Himawari-8 TSS at 10-min temporal frequency.

Continuous $b_{bp}$ (595 nm) data were acquired at the LJCO with the Sea-Bird Scientific ECO-BB9 instrument deployed at 3 m depth. The $b_{bp}$ measurements were acquired following equipment cleaning and redeployment to avoid potential contamination due to bio-fouling of the BB9 instrument. As a result, the time difference between the closest available $b_{bp}$ to the gravimetric TSS sample was variable but mostly within 1 h. Linear regression between near-concurrent pairs of gravimetric TSS and the closest available $b_{bp}$ (595 nm) was derived based on Soja-Woźniak et al. [42] (Figure 4). The in-situ measurements of TSS ranging from 1.3 to 22.8 mg L$^{-1}$ yielded a positive correlation (R$^2$ = 0.79) to $b_{bp}$ between 0.1 and 0.15 m$^{-1}$. The $b_{bp}$ versus TSS relationship calculated in this study was comparable to the relationships previously calculated for the GBR waters [41,42].

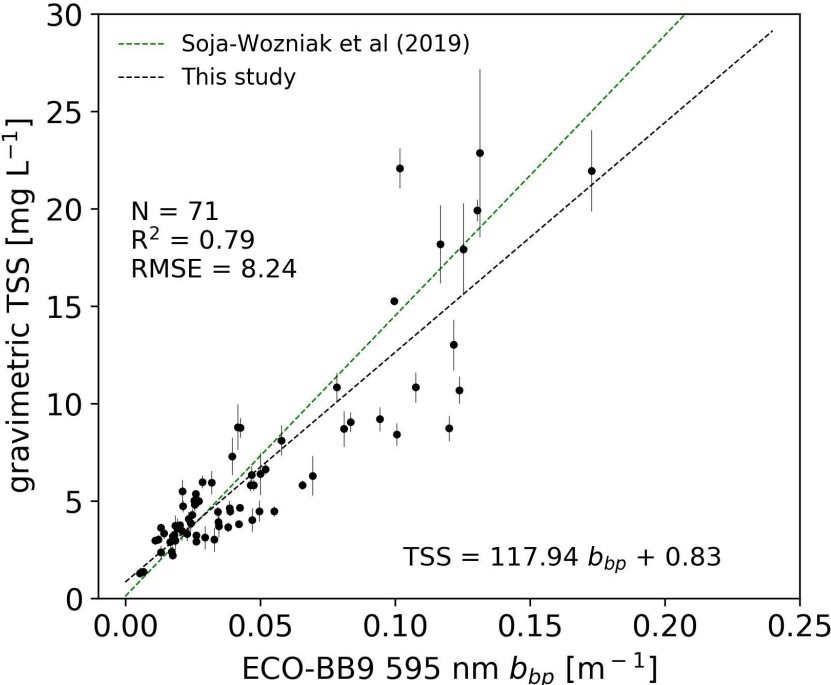

**Figure 4.** The linear relationship between gravimetric Total Suspended Solids (TSS) and nearly-concurrent particulate backscattering ($b_{bp}$ at 595 nm) data measured at the Lucinda Jetty Coastal Observatory (LJCO). The dashed lines represent the relationship computed in the present study (black) and in Soja-Woźniak et al. [42] (green). The error bars represent the standard deviations of in situ triplicate gravimetric TSS.

For instance, the Soja-Woźniak et al. [42] relationship was calculated from 34 pairs of in situ TSS and daily aggregated measurements of $b_{bp}$ (595 nm) at the LJCO. The largest fraction of particles found at LJCO were of inorganic origin, characteristic of estuarine and river runoff dominated waters. The reviewed relationship was applied to the ECO-BB9 $b_{bp}$ (595 nm) measurements aggregated into 10-min averages for deriving in situ TSS (hereafter $TSS_{b_{bp}}$) and for a systematic comparison against concurrent Himawari-8-derived TSS (hereafter $TSS_{H8}$).

### 2.4. Total Suspended Solids Retrievals from Himawari-8 AHI

The Himawari-8 AHI TSS algorithm applied in this study was developed based on inverse modelling of coupled ocean-atmosphere radiative transfer simulations using Artificial Neural Networks (ANN) [78]. The radiative transfer bio-optical model parameterizations follow an approach previously developed for European coastal waters [79–81] that was adapted to the regional optical conditions of the GBR lagoon using in situ data [57,59]. The Neural Network algorithm was trained with the top of atmosphere simulated AHI radiance data for a wide range of atmospheric and oceanic conditions to directly estimate the concentration of TSS. The derived Himawari-8 AHI TSS outputs, hereafter $TSS_{H8}$, were hourly aggregated and extracted as the median value of a 3-by-3-pixel subset centred at the location of the in situ data points, complying with ocean colour validation protocols [82]. The $TSS_{H8}$ were validated with independent in situ TSS (N = 347) data collected in the GBR region between 2014 and 2018, with a third of the dataset obtained from the LJCO. Validation with in situ observations showed that the algorithm is capable of estimating TSS between 0.25 and 100 mg L$^{-1}$ with relative errors within 75% and detection limits (0.25 mg L$^{-1}$) comparable to those achieved for the in situ gravimetric method (0.4 mg L$^{-1}$) [83]. An in-depth description of the validation procedure is available in Patricio-Valerio [59] and Patricio-Valerio et al. [57]. The algorithm was applied to 10-min Himawari-8 observations, which were quality controlled for the subsequent time series analysis by masking clouds [84], land, sun glint, and out-of-range values, following the processing steps described in Patricio-Valerio et al. [57] and in Patricio-Valerio [59].

### 2.5. Time Series of Himawari-8 and In Situ TSS

The median and standard deviation of all valid Himawari-8 AHI pixels within a 3-by-3-pixel subset centred at the LJCO coordinates (18.5°S, 146.4°E) were extracted from each 10 min $TSS_{H8}$ product. The 3-by-3-pixel window was chosen instead of a single pixel as a reasonable sampling strategy to increase the likelihood of obtaining a valid observation while avoiding data gaps due to passing clouds. Likewise, a temporal rolling median with a window of 20 min was applied to the daily $TSS_{H8}$ time series to fill in data gaps without overly smoothing TSS fluctuations. Time series of concurrent $TSS_{b_{bp}}$ and of $TSS_{H8}$ were extracted on the 25th and 26th of April (namely April #1 and April #2, respectively) and on the 11th and 12th of May (namely May #1 and May #2, respectively) of 2018. These dates were chosen based on the availability of matching in situ and satellite data and observable plume dynamics verified through supplementary webcam imagery at the LJCO. A minimum of 4 h of cloud-free $TSS_{H8}$ products were required each day for a comprehensive analysis of diurnal variability within a semi-diurnal tidal cycle [53]. The $TSS_{H8}$ and $TSS_{b_{bp}}$ time series were visually compared, and their local maxima were computed for the assessment of temporal coherency. The difference between minimum and maximum TSS values across a day of observations ($\Delta$TSS in mg L$^{-1}$) was calculated for $TSS_{b_{bp}}$ and $TSS_{H8}$ values as a proxy of diurnal variability. In addition, peak TSS values ($P(TSS_{b_{bp}})$ and $P(TSS_{H8})$) were indicated in time series with a larger marker and values compiled in the table for reference.

The concentration of non-algal particulate matter (NAP in mg L$^{-1}$) derived from daily MODIS-Aqua observations (MODISA-NAP) was utilized as a proxy of TSS and added to the time series plots for inter-comparison. The MODISA-NAP products were derived from regional coastal ocean colour algorithms [85–87] developed by CSIRO for the GBR region

under the eReefs project [88]. Ongoing development of TSS products from OLCI and VIIRS sensors needs validation for the GBR [52] and is not yet available for this analysis. Therefore, the overpass times of Sentinel-3A (S3A) OLCI and VIIRS (SNPP and JPSS) satellite sensors were extracted [89,90] and overlayed in time series to show the availability of matching observations provided by LEO ocean colour sensors over LJCO, given cloud-free data would be available. The overpass of the Sentinel-3B OLCI sensor was not included in the time series due to its launch on 25 April 2018, and matching imagery to April and May time series was not yet available.

The central-north GBR, where LJCO is located, received above-average rainfalls during the 2017-2018 wet season, largely a result of a low-pressure system and the subsequent passage of a tropical cyclone Nora in late March 2018 [91]. Nora approached the Gulf of Carpentaria to make landfall on the west coast of the Cape York peninsula as a category 3 system on the 24th of March. Nora delivered extreme rainfalls (~800 mm/month), and flash floods were experienced. The Herbert River discharge peaked at 500,000 ML/day in mid-March [63], marking the 4th major discharge event since 2010 (Figure 5).

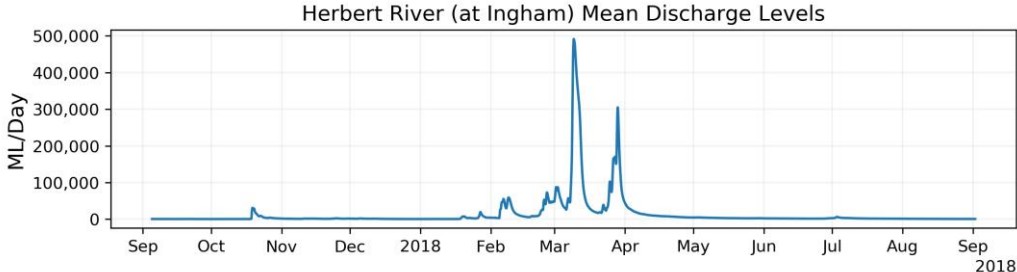

**Figure 5.** Herbert River daily mean discharge levels in Megaliters per day (ML/day) between September 2017 and September 2018. Data were acquired from the Herbert River gauge station at Ingham (Site n° 116001F at 18.63°S, 146.14°E), distant 30.5 km from the stream mouth [79].

The April and May time series were recorded a few weeks after the low-pressure system passed through the Herbert River catchment area. In early April up to 50,000 ML were discharged per day, indicating a residual flow of freshwater from the preceding weeks of intense rainfall. However, discharge levels dropped below 6000 ML/day in late April and early May 2018, to be comparable with discharge recoded in previous dry seasons.

Above-water webcam imagery at the LJCO facility (facing Hinchinbrook Island) was inspected to aid the interpretation of the time series. The 5-min interval snapshots show the arrival of riverine plume waters at the LJCO for each selected date (Figure 6). The median and standard deviation $TSS_{H8}$ values of a 3-by-3-pixel box centered at this site are annotated on the images for reference. The timing of plume arrival, as identified by visual inspection of the webcam images (Figure 6), was annotated in each time series plot for comparison. In addition, the hourly tidal height measured at the Lucinda jetty by the Queensland Government [63] was overlaid to the time series to investigate the potential of tidal influences on TSS fluctuations. In addition, the maps of $TSS_{H8}$ centered at the site coordinates were derived at an hourly temporal resolution to help with the interpretation of the spatial variability of the plume.

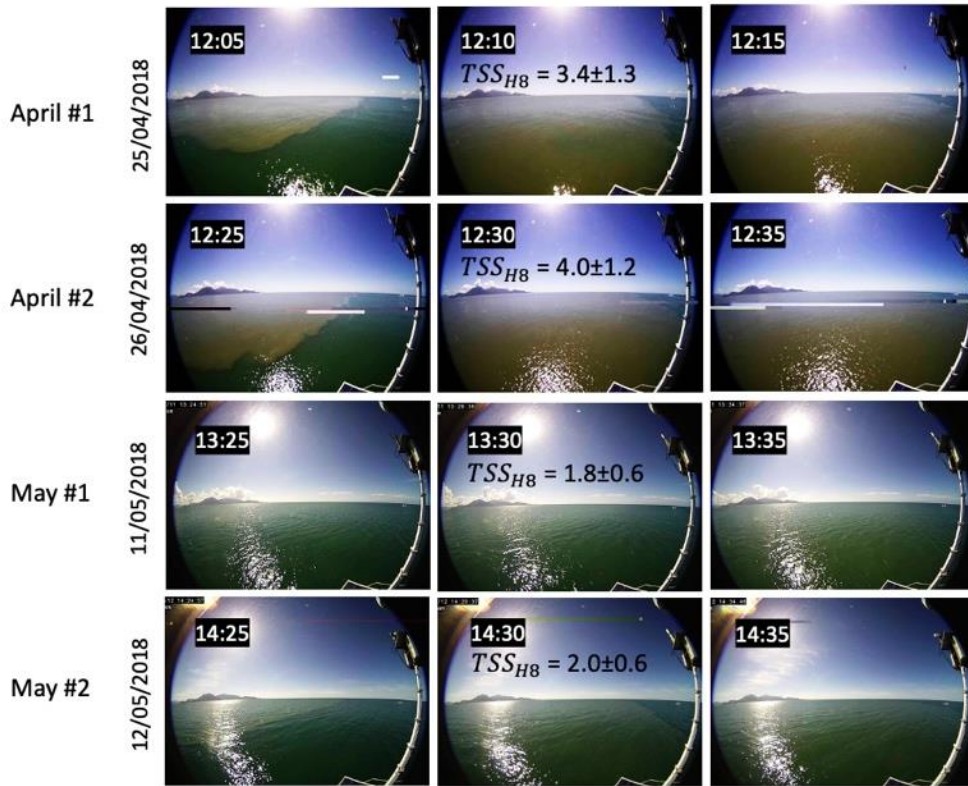

**Figure 6.** Webcam images of the Herbert River plume arriving at the LJCO during April and May selected time series. Hinchinbrook Island is visible on the left edge of the pictures. The corresponding median and standard deviation of $TSS_{H8}$ (mg L$^{-1}$) is annotated for the time of the plume's arrival at LJCO.

### 2.6. Monthly Maximum Diurnal Variability of TSS

In addition to the time-series analysis, we investigated the spatio-temporal patterns of episodic events contributing to the water quality dynamics at LJCO based on one year of Himawari-8 TSS 10-min products. Himawari-8 observations acquired between 8 am and 4 pm and between August 2017 and July 2018 were extracted for the LJCO area and processed to TSS products according to the steps described in Patricio-Valerio et al. [57]. The 10-min observations were hourly aggregated to speed up the processing and to provide useful products at a reporting level, resulting in up to 8-hourly TSS products daily. A minimum of 4-hourly TSS products a day were required to resolve diurnal variations [53]. Hourly masks for land, clouds, and sun glint were applied to the corresponding TSS products. Likewise, out-of-range input and output values were masked in the Himawari-8 data using the associated ANN algorithm flags computed after the inversion procedure, following the methods described in Patricio-Valerio et al. [57] and in Patricio-Valerio [59]. Diurnal TSS features were determined by calculating the absolute difference between the minimum and maximum TSS within a day ($\Delta$TSS), pixel-by-pixel, and then the maximum $\Delta$TSS value within a given month (Max$_{\Delta TSS}$). The Max$_{\Delta TSS}$ was then utilized to investigate the occurrence of episodic events or 'hotspots' of diurnal changes in TSS that may contribute to GBR water quality dynamics. Here, the term 'hotspot' was defined as water pixels with Max$_{\Delta TSS} \geq$ ~5 mg L$^{-1}$, indicating a potential exceedance of guideline values in open coastal and midshelf waters. Max$_{\Delta TSS}$ values below 0.25 mg L$^{-1}$ were masked out to comply with the detection limits of the algorithm [57].

The spatial distribution of Max$_{\Delta TSS}$ (Figure 7a) computed for September 2017 reveals hotspots of TSS at both outlets of the Hinchinbrook Channel and suggests areas where episodic events, such as floods, may have occurred. In contrast, the spatial distribution of monthly median TSS (Figure 7b) only emphasizes the northern branch of the plume,

with TSS consistently above ~3 mg L$^{-1}$. Therefore, the Max$_{\Delta TSS}$ maps allowed us to identify locations with extreme and episodic short-term diurnal fluctuations that would have been overlooked by utilising weekly to monthly averages or median composites of TSS (Figure 7b). The monthly Max$_{\Delta TSS}$ maps were grouped into the wet and dry seasons to facilitate the identification of seasonal patterns.

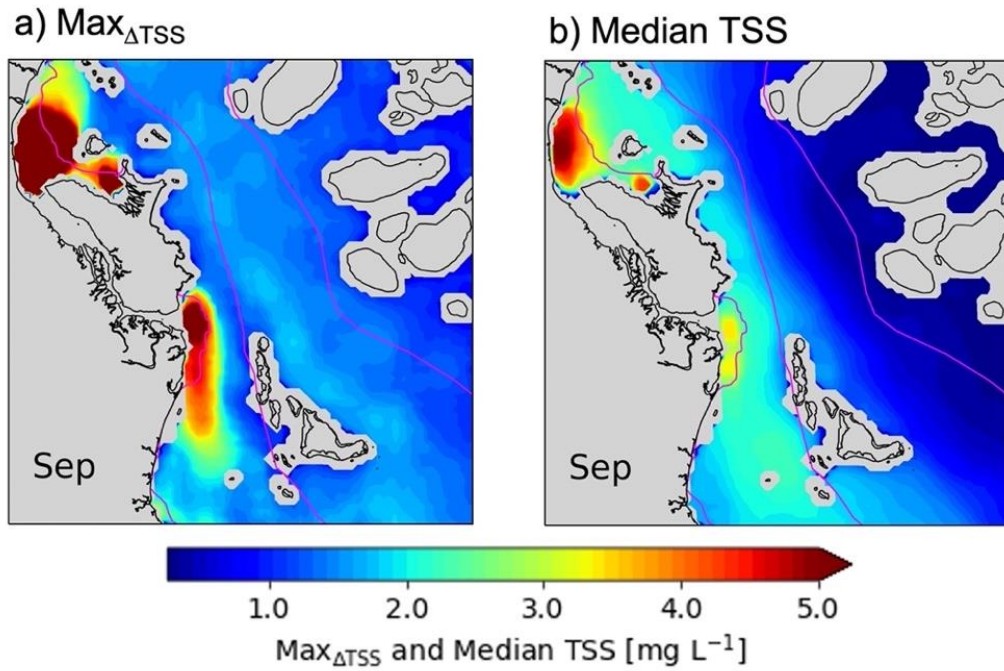

**Figure 7.** Comparison between monthly Max$_{\Delta TSS}$ (**a**) and median TSS values (**b**) at LJCO and in the GBR lagoon for September 2017.

### 2.7. *Frequency Exceedance of TSS above GBR Guideline Thresholds*

We computed a frequency exceedance map to illustrate the utility of Himawari-8 in providing an objective water quality management tool for the GBR with minimum data-gaps. The frequency exceedance was computed as the percentage of time for which TSS exceeded the GBR annual water quality guideline threshold of 2 mg L$^{-1}$ for open coastal and mid-shelf waters [73]. Hourly maps of Himawari-8 TSS with appropriate masks were generated for the period between August 2017 to July 2018, according to Patricio-Valerio et al. [57]. The total number of times a pixel exceeded TSS > 2 mg L$^{-1}$ was recorded (total counts), as well as the total number of valid observations (i.e., not masked for clouds, glint, or out-of-range values) within the same timeframe. The frequency exceedance was calculated as the ratio between the total counts and the total valid observations and multiplied by 100%. Polygons delineating the Marine Water Bodies around the LJCO were obtained from the Great Barrier Reef Marine Park Authority (GBRMPA) database and overlaid to the exposure map to help the identification of the open and enclosed coastal waters.

## 3. Results

The webcam images of Figure 6 revealed that the plume front, indicated by the contrast between brown and green waters, passed the LJCO site within 10 min. The colour changes between the brown plume and the green coastal waters were clear in the April images and visible but more subtle in the May images. Both April and May time series were recorded during neap tides, only two weeks apart, with April showing relatively larger tidal ranges (Table 2). The webcam images at the LJCO (Figure 6) also showed the timing when the Herbert River plume reached the facility, indicated by a vertical grey shading in the time

series of Figure 8. The webcam-derived timing for the onset of the plume reasonably matched the $P(\text{TSS}_{b_{bp}})$ and $P(\text{TSS}_{H8})$, generally within 40–30 min (except for May #2).

**Table 2.** Summary data for TSS time series: values of ΔTSS from backscattering data and Himawari-8, respective peak values, and intra-pixel standard deviations (SD); time difference between TSS peak values ($\Delta t$); tidal ranges; ΔTSS and maximum values with units in mg L$^{-1}$.

|  | **April #1** | **April #2** | **May #1** | **May #2** |
|---|---|---|---|---|
| $\Delta TSS_{b_{bp}}$ | 6.1 | 4.2 | 1.4 | 2.0 |
| $\Delta TSS_{H8}$ | 7.3 | 6.7 | 1.2 | 1.5 |
| $P(TSS_{b_{bp}})$ | 8.8 | 8.1 | 3.4 | 4.2 |
| $P(TSS_{H8})$ (SD) | 9.8 ($\pm$3) | 9.0 ($\pm$4) | 3.0 ($\pm$1) | 2.5 ($\pm$0.5) |
| $\Delta t$ (hours: minutes) | 00:10 | 01:10 | 00:40 | 01:00 |
| Tidal range (m) | 2.5 | 2.7 | 2.3 | 2.0 |

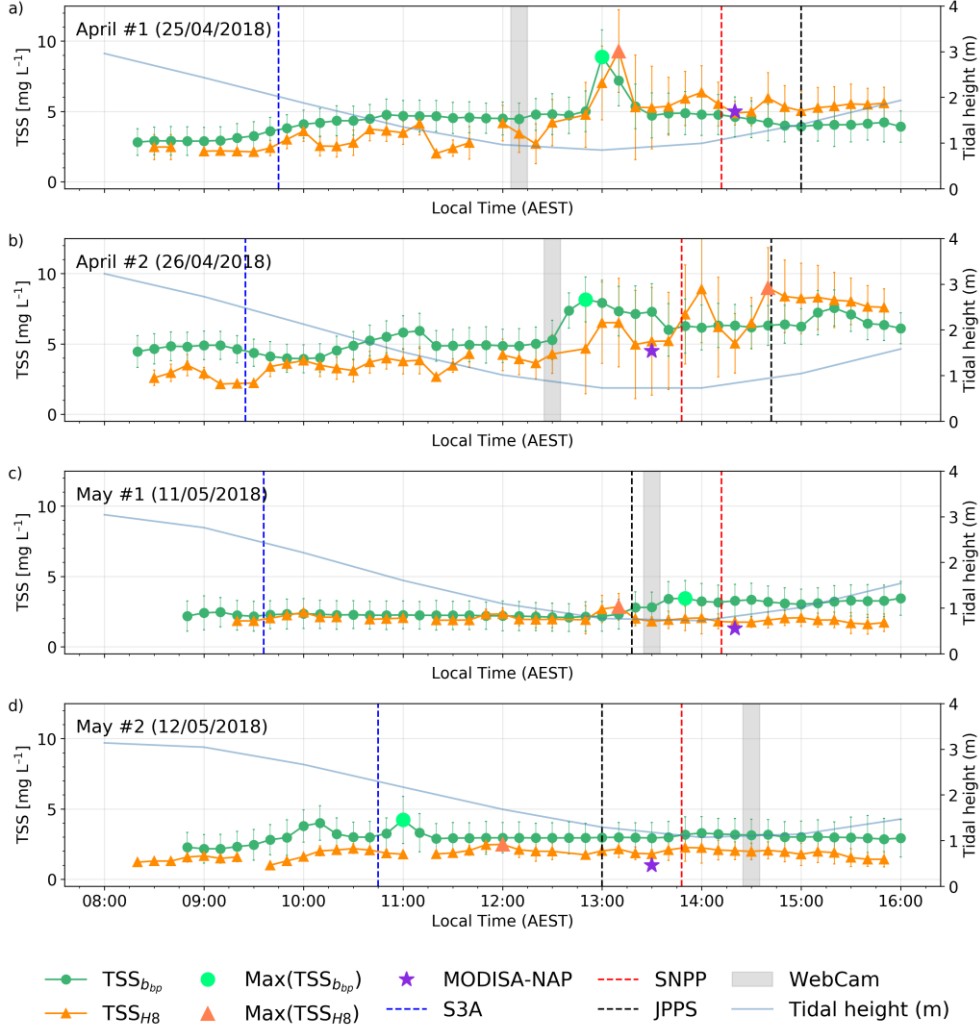

**Figure 8.** Time series of 10 min TSS$_{H8}$ (dark orange) and TSS$_{b_{bp}}$ (green) for April (**a**,**b**) and May (**c**,**d**) 2018 at the LJCO. Peak TSS are annotated with large markers. The MODISA-NAP value (purple star), and Sentinel-3A OLCI, SNPP, and JPPS VIIRS overpasses are annotated as vertical dashed lines. The time of plume arrival derived from webcam observations (grey shading) and the hourly tidal height (light blue) were included. Error bars represent the standard deviation of each TSS measurement.

$TSS_{H8}$ was generally underestimated and moderately correlated with $TSS_{b_{bp}}$ ($R^2 < 0.49$). However, the small relative differences (RMSE < 1.7 mg L$^{-1}$) and absolute percentage differences (MAPE < 38%) indicate that $TSS_{H8}$ values were, in most cases, reasonably comparable to $TSS_{b_{bp}}$. The temporal patterns of $TSS_{H8}$ and $TSS_{b_{bp}}$ closely matched (Figure 8), with moderate to strong diurnal fluctuations (1.2 to 7.3 mg L$^{-1}$) observed within 8 h (Table 2). The $\Delta t$ were mostly within 1 h, with the April #1 time series exhibiting the closest temporal match between $TSS_{H8}$ and $TSS_{b_{bp}}$ ($\Delta t$ = 10 min). The intra-pixel standard deviations (SD) computed for $TSS_{H8}$ showed the largest values (~4 mg L$^{-1}$) occurring around the time of plume arrival, particularly for the April time series (Figure 8a,b).

Strong diurnal variability of satellite and in situ TSS (~4 to 6 mg L$^{-1}$) were observed in both April time series, with $\Delta TSS_{H8}$ consistently higher than $\Delta TSS_{b_{bp}}$. In both April scenarios, the TSS reached its magnitude peak during the slack tides between 12 and 2 pm. Meanwhile, moderate diurnal variability (<~2 mg L$^{-1}$) was observed in the May time series (Figure 8c,d), with May #1 peak TSS occurring during the slack tide, consistent with the April time series. In May #2, however, both in situ and satellite peak TSS occurred 2–3 h before the slack tide and before the plume arrival recorded by the webcam.

The marked overpasses of LEO ocean colour satellites indicate that at least one morning (S3A) and three-afternoon observations (MODIS-Aqua, SNPP, and JPPS VIIRS) would be available for diurnal inference. However, these would likely miss the peak of the plume at LJCO, particularly if that happens between 11 am and 1 pm. The overpass of S3A at LJCO on April #1 and May #1 time series (blue dashed line) occurred at the start of the ebbing tide, hours before the plume arrival and TSS peaks at LJCO. Meanwhile, the overpasses of SNPP and JPPS satellites, as well as the MODIS-Aqua, matched the low tide in the afternoon but often missed the peak of TSS with the plume's arrival. On April #2, however, the four LEO satellite sensors would have reasonably captured the diurnal changes, with S3A at high tide and low TSS values, and MODIS-A, JPPS, and SNPP observations at low tide and matching the rapid changes in TSS observed by Himawari-8. Yet, in this scenario, the LEO satellites would have missed the significant variations recorded between 12:15 pm and 1 pm by the in-situ probes and webcam. The MODISA-NAP ($M_{NAP}$) values closely matched the concurrent $TSS_{H8}$ in all scenarios investigated, with a root mean squared error (RMSE) of 0.57 mg L$^{-1}$ and a mean absolute percentage error (MAPE) of 32%.

Hourly TSS maps from Himawari-8 (Figure 9) computed for the April #2 and May #2 time series illustrate the diurnal spatio-temporal variability of the plume and the heterogeneous distribution of TSS in the GBR lagoon. The Herbert River plume was characterised by orange-red areas with TSS $\geq$ ~5 mg L$^{-1}$ and visually distinctive from the surrounding coastal waters.

In April #2, most of the area centred at the LJCO (indicated by a cross marker) was covered by plume waters as the tide retreated (ebbing from 8 am to 2 pm). However, the plume identified in May #2 was notably smaller, and its waters seldom reached the site. In fact, subtle TSS fluctuations ($\Delta TSS_{b_{bp}} < 2$ mg L$^{-1}$) occurred throughout the day for May #2 time series, despite marked changes in water colour observed from the webcam images (Figure 6). Nevertheless, is it possible that the perceived water colour changes, as seen from the webcam at 2:30 pm, were caused by riverine waters with TSS < 2 mg L$^{-1}$, which were not detectable from the hourly TSS maps at 1 km spatial resolution.

Hotspots of Max$_{\Delta TSS}$ were investigated at LJCO and in adjacent coastal areas for the typical wet and dry season months between 2017 and 2018 (Figure 10). Large diurnal changes ($\geq 5$ mg L$^{-1}$) persisted during the months investigated and marked seasonal patterns were not detected. The southern and northern branches of the Hinchinbrook Channel were clearly delineated by Max$_{\Delta TSS}$ above ~5 mg L$^{-1}$ across all months, except for the southern branch in October, where Max$_{\Delta TSS}$ only reach about ~2 mg L$^{-1}$. During the start of the wet season (November–January), the southern TSS hotspot was usually restricted around the channel outlet, forming a distinctive estuarine plume. However, as the wet season developed (February–May), the southern branch of the TSS hotspot connected itself to waters from the southern coastal areas. In March, following major wet season flood

discharges (~500,000 ML/day, Figure 5), the TSS hotspots extended further offshore. In May, a contiguous coastal feature of moderate-high ($\geq$~3 mg L$^{-1}$) Max$_{\Delta TSS}$ was delineated between the enclosed coastal waters and the mid-shelf lagoon. From June to September, diurnal TSS changes above ~1 mg L$^{-1}$ extended further offshore. Additionally, a discrete Max$_{\Delta TSS}$ feature developed south of the LJCO site between August and September.

Figure 11 illustrates the monthly frequency exceedance maps computed from Himawari-8 observations acquired from August 2017 to July 2018. In general, TSS exceeded guideline thresholds in the open coastal and midshelf waters of the GBR during the entire observational period evaluated and were high over large areas, especially from March to September. Exceedances were recorded for at least 60% of the total valid observations (Figure 12) located within the open coastal waters.

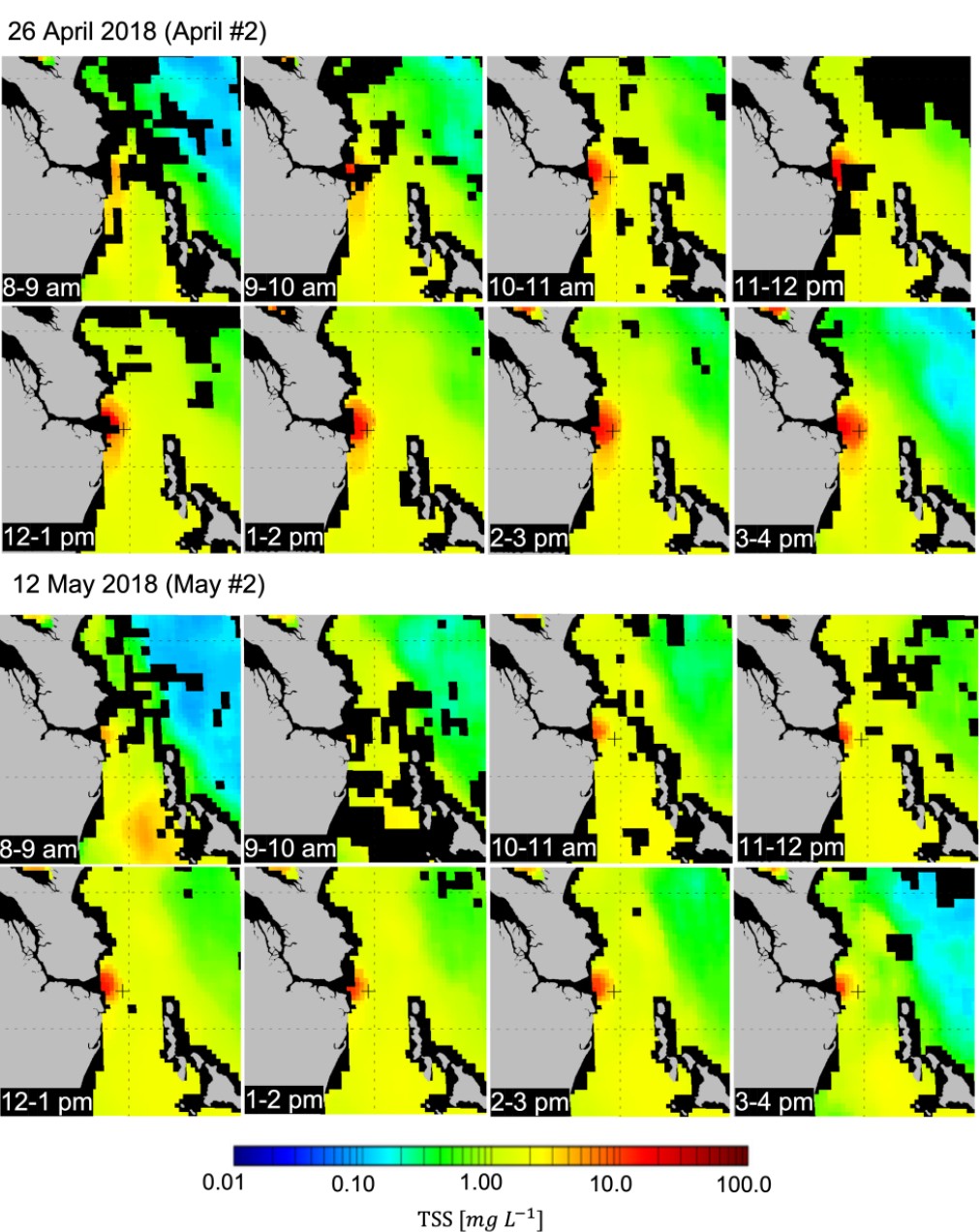

**Figure 9.** Hourly Himawari-8 TSS over the LJCO from 8 am to 4 pm local time (AEST) on 26 April 2018 (April #2, **top** panel) and on 12 May 2018 (May #2, **bottom** panel). Emerged surfaces are masked in grey, while clouds and nearshore areas are masked in black. The cross marker indicates the location of the LJCO facility. The time range annotated in each plot refers to the interval of observations utilized for hourly aggregation of Himawari-8 observations.

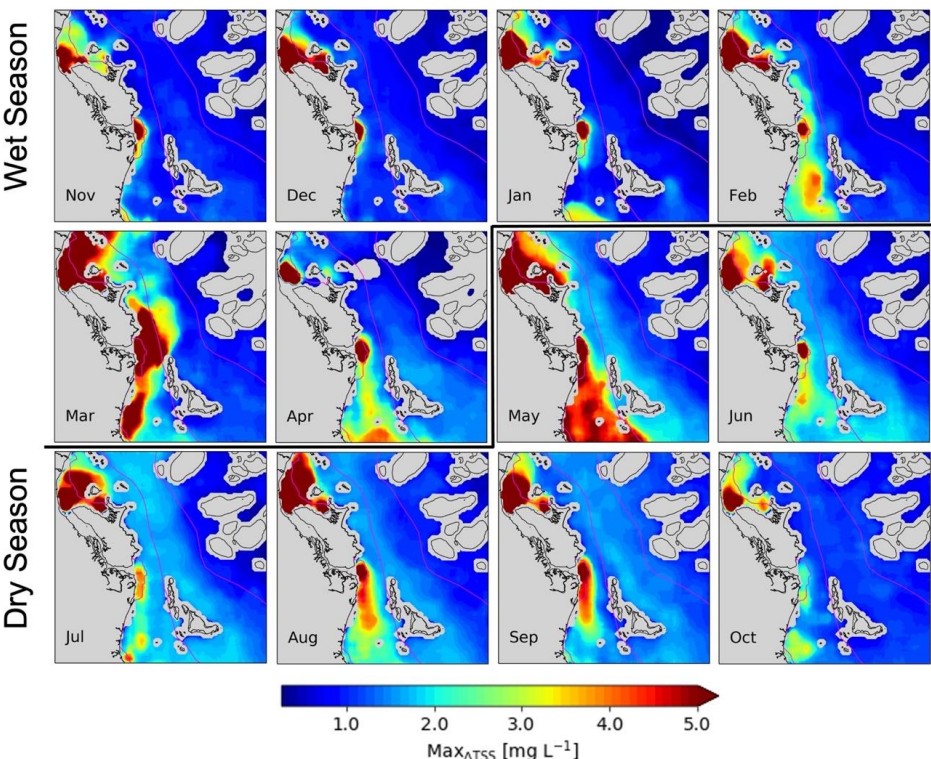

**Figure 10.** Max$_{\Delta TSS}$ at the LJCO for wet and dry seasons between 2017 and 2018. Land, shallow waters and reef areas, unavailable observations, and Max$_{\Delta TSS}$ lower than 0.25 mg L$^{-1}$ are masked in grey. The delineation of the Marine Water Bodies (enclosed, open coastal, mid-shelf, and offshore) is represented with magenta contour lines.

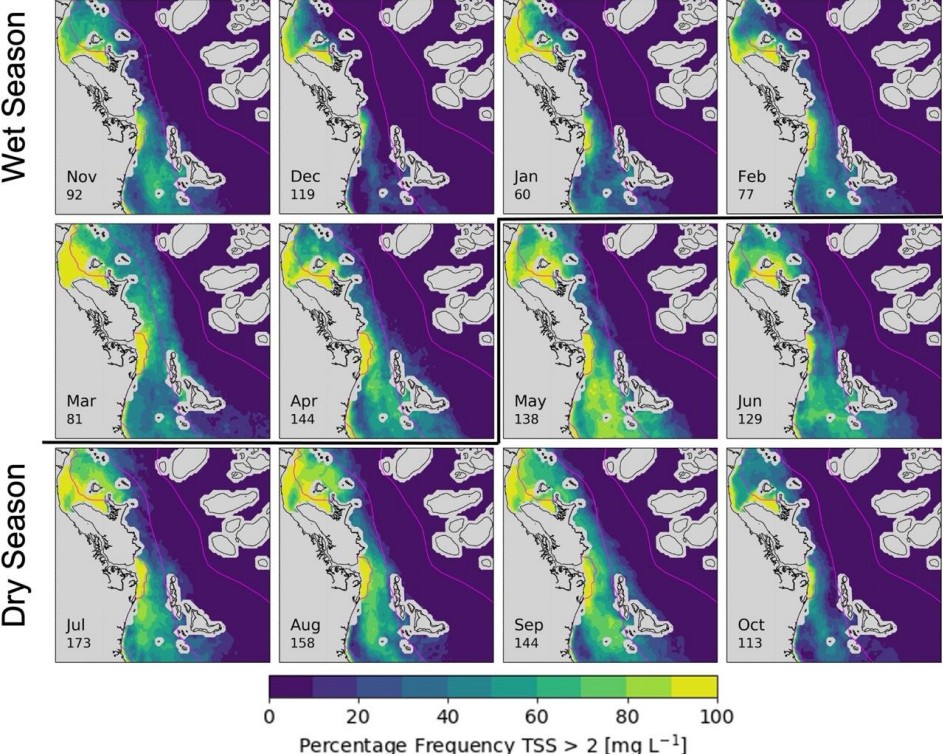

**Figure 11.** Percentage frequency where TSS > 2 mg L$^{-1}$ between 2017 and 2018. Land, shallow waters, and reef areas are masked in grey. The delineation of the Marine Water Bodies (enclosed, open coastal, mid-shelf, and offshore) is represented with magenta contour lines. The maximum number of valid observations recorded is annotated on each map.

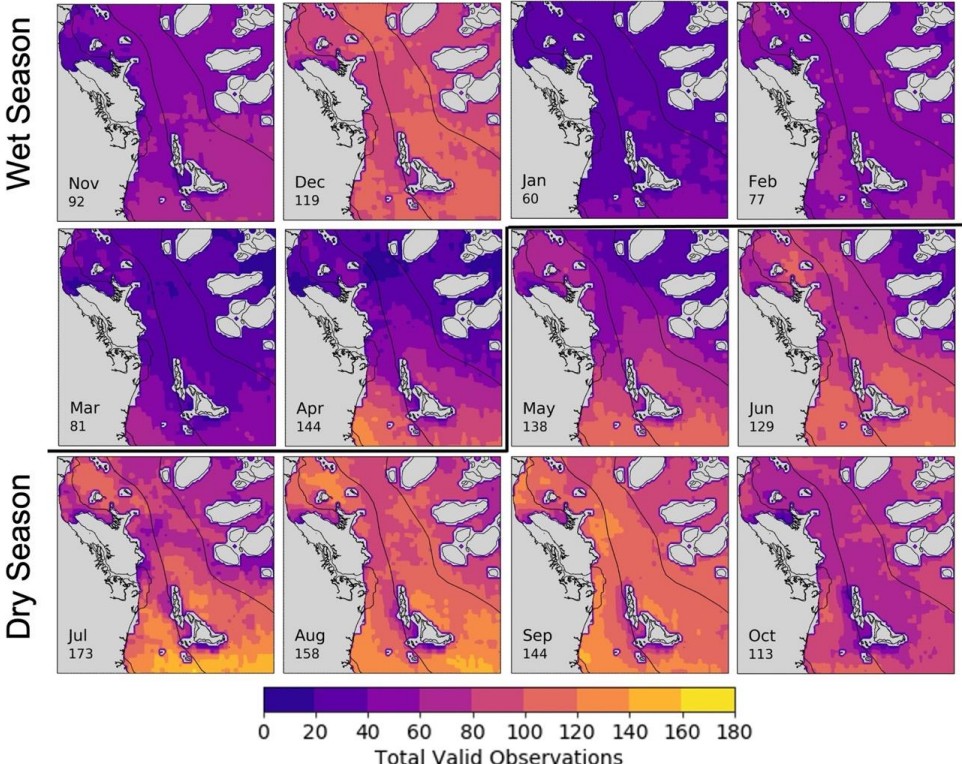

**Figure 12.** Total valid observations computed by pixel. A valid observation is free from clouds, sun glint masking, and out-of-range ANN flags. Land and reef areas are masked in grey. The delineation of the Marine Water Bodies (enclosed, open coastal, mid-shelf, and offshore) is represented with contour lines. The maximum number of valid observations recorded is annotated on each map.

The TSS guideline thresholds were exceeded less often (<60% of observations) between December and February, and in October, particularly for the mid-shelf waters. This exposure area corroborates the patterns found in Figure 10, where hotspots of $Max_{\Delta TSS} \geq 5$ mg $L^{-1}$ was identified in the open coastal waters around LJCO and on both branches of the Hinchinbrook channel. The reef areas offshore were seldom impacted (<10%) by TSS concentrations above guideline thresholds, except the northeast of Hinchinbrook Island during March 2018. The number of valid observations (i.e., not masked by clouds, sun glint, and out-of-range input and output values) per pixel is illustrated in Figure 12 to help interpretation of Figure 11 maps. The areas with fewer observations (<20) were consistently located adjacent to coastal waters, islands, and reefs offshore, particularly between January to April, when clouds and the sun glint disk persist over the area [59].

A maximum of 173 valid observations were available for computation in July, an average of ~5 per day, whilst 60 to ~80 observations per pixel were available for each month between January to March, an average of ~2 to 3 per day. Although the number of valid observations for the computation of frequency maps may be reduced from January to March, the exceedances recorded were significant (>60%) for the open coastal waters. In contrast, around 100 valid observations were captured during December, when the frequency of TSS values above the guideline threshold for open coastal waters was low (<40%) and for the midshelf and offshore areas negligible (<10%).

## 4. Discussion

Diurnal fluctuations of TSS were investigated utilising concurrent in situ and Himawari-8 AHI satellite data at the LJCO. The continuous measurements of particulate backscattering and its computed TSS ($TSS_{b_{bp}}$), supported a comparison with the 10-min Himawari-8-derived TSS ($TSS_{H8}$) for an integrated assessment of great temporal changes in water quality. Overall, $TSS_{H8}$ agreed remarkably well (MAPE within 38%) with the concurrent

$TSS_{b_{bp}}$ at the LJCO (Figure 8), despite methodological differences. $TSS_{H8}$ fluctuations of ~7 mg L$^{-1}$ were observed within a day at the LJCO, corroborating previous studies employing geostationary observations for diurnal water quality assessments [92–95].

The observed TSS fluctuations during two days in April and May 2018 were associated with the Herbert River discharges and with the local tidal dynamics. In most cases, the plume reached the LJCO at slack tide, followed by rapid (<1 h) and significant increases of TSS up to 7.3 mg L$^{-1}$. These results are supported by previous findings, suggesting a persistent semi-diurnal tidal influence on the bio-optical variability at LJCO [42]. However, the sharp contrast between river plume and coastal waters (Figure 9), as well as the peak TSS concentrations (~9 mg L$^{-1}$), indicate that the April plume waters were largely influenced by the residual volume of freshwater discharged from the prevailing wet season. The high volume discharged from the Herbert River prior to April 2018 (~500,000 ML/day) may have superimposed the tidal influence at the LJCO by increasing the horizontal advection and vertical mixing and, thus, modulating significant diurnal fluctuations of TSS. Moreover, the diurnal fluctuations of TSS presented in this study were recorded during neap tides on the first and third quarter moon, with ranges of up to 2.7 m. Therefore, diurnal fluctuations of TSS > 7 mg L$^{-1}$ may be experienced during spring tides at this site, with tidal ranges larger than 3 m and associated with larger exchanges of water volume, likely increasing TSS advection [96].

The 10-min TSS and hourly maps from Himawari-8 were useful and complementary tools to investigate diurnal variability in the coastal GBR. The 10-min observations from Himawari-8 were appropriate to track rapid (<1 h) and moderate (>1 mg L$^{-1}$) fluctuations in turbid coastal waters with the time series analysis. Meanwhile, the hourly Himawari-8 derived TSS provided sufficient resolution to successfully map diurnal variability and track river plumes in the coastal GBR. The MODISA-NAP products yielded comparable results to concurrent $TSS_{H8}$, with minor discrepancies possibly associated with algorithm parameterization and atmospheric correction of MODIS-Aqua observations, as reported in Dorji and Fearns [97]. A comprehensive inter-comparison between concurrent MODIS-Aqua, Sentinel-3 OLCI, VIIRS-JPSS, -SNPP, and Himawari-8 AHI-derived TSS products should provide more understanding of their methodological differences and limitations. However, while all of these sensors combined may provide morning and afternoon observations [98,99], these would still be insufficient to fully depict the local bio-optical diurnal variability, as represented by Himawari-8-derived TSS every 10 min.

Although Himawari-8 AHI provided an unprecedented amount of data for water quality observations in the coastal GBR, $TSS_{H8}$ presented a temporal mismatch and a systematic underestimation compared to $TSS_{b_{bp}}$ due to the much larger spatial representation and aggregation of the satellite data. The differences in magnitude between $TSS_{H8}$ and $TSS_{b_{bp}}$ could also be attributed to the current limitations of the present algorithm (as discussed in Patricio-Valerio et al. [57]) and from the limitations of the in situ data in utilising $b_{bp}$ data as a proxy for TSS. At LJCO, the suspended solids are mainly a mix of white organic marine carbonate sands and brown inorganic terrigenous muds (see Figure 2), each of which exhibits a distinctive relationship against $b_{bp}$ [42]. As a result, the functional relationship between $b_{bp}$ and gravimetric TSS may result in several distinctive slopes contributing to a larger scatter and uncertainty of this functional relationship (see Figure 4) depending on the proportion of organic to inorganic material in the dataset utilized. In addition, the accuracy with which TSS can be derived from $b_{bp}$ in highly dynamic coastal waters is possibly limited by the time difference between each pair of measurements. As a result, derived TSS can be over or underestimated, depending on the stability of local particle composition to different tidal stages. Further improvements could include limiting the time difference between measurements of $b_{bp}$ and TSS to within one hour or less to avoid a mismatch due to sample collection at different tidal stages.

Additionally, it was recognised that the $TSS_{b_{bp}}$ response was delayed compared to the time the plume reached the LJCO, as recorded on the webcam images. This temporal lag may be associated with a strong vertical stratification due to salinity differences induced

by the plume waters while moving cross-shelf [70]. The buoyant plume of freshwater may be restricted to the surface as a wedge when reaching the site. Consequently, the $b_{bp}$ sensor deployed at 3 m depth miss the arrival of the plume waters at the surface until more vigorous vertical mixing takes place, causing a sharp gradient (generally larger than 2 mg L$^{-1}$) in the $b_{bp}$ and derived TSS. Although sub-surface $b_{bp}$ measurements compare well to satellite-derived products in open ocean waters [100], it may be invalid to assume this applies in optically complex coastal waters. Alternative bio-optical measurements, such as absorption and turbidity [94], can be explored and provide more clarity on understanding the timing of plume arrival and the relationship to Himawari-8 derived TSS.

Nevertheless, the present method's main limitations were due to the low sensitivity of Himawari-8 AHI visible and near-infrared bands to small changes in ocean colour. A sensitivity analysis suggested that Himawari-8 TSS within 0.25 mg L$^{-1}$ were likely below the detection limits of the algorithm employed here, producing errors with up to three orders of magnitude difference [57]. Thus, TSS$_{b_{bp}}$ fluctuations lower than 0.25 mg L$^{-1}$ were not well discriminated by TSS$_{H8}$ on some occasions, which was illustrated in the time series extracted on May #1 and May #2.

The temporal difference between peaks of TSS$_{H8}$ and TSS$_{b_{bp}}$ ranged between 10 min to 1 h. Likewise, Neukermans et al. [94] observed good correspondence between the SEVERI-derived TSS and in situ diurnal turbidity variations, with an average temporal lag of 11 min between peak values. The temporal difference between TSS$_{H8}$ and TSS$_{b_{bp}}$ peak values may be explained by the spatial mismatch between the in-situ point sample, represented by the TSS$_{b_{bp}}$, and the 3-by-3 pixel-box (9 km$^2$) representing TSS$_{H8}$. In fact, TSS$_{H8}$ derived from the spatial median of 9 pixels generally presented increased intra-pixel standard deviations with rapid increases of TSS, indicating intra-pixel heterogeneity likely triggered by the plume arrival. It is acknowledged that the TSS retrievals are largely impacted by the nominal spatial resolution utilized [101], particularly in highly dynamic coastal waters. The Himawari-8 AHI red band centred at 640 nm with an original spatial resolution of 0.5 km, rather than resampled to 1 km, may be sought for improved TSS retrievals in the coastal GBR.

The maximum diurnal variability of TSS, Max$_{\Delta TSS}$, was computed to explore the spatial extent of TSS hotspots in the GBR during dry and wet season months. A TSS hotspot was defined as the area adjacent to the coast where the Max$_{\Delta TSS}$ was higher than ~5 mg L$^{-1}$, which was visually detected as a sharp colour contrast to lower Max$_{\Delta TSS}$. The TSS hotspots characterised waters where episodic variations potentially exceeded the annually averaged threshold values for TSS in the open coastal and midshelf areas. Consequently, TSS hotspots may indicate areas where TSS concentrations exceed the suggested guideline thresholds in enclosed coastal waters. The TSS hotspots are largely associated with short-lived coastal processes known for mobilising TSS, such as tidal currents and jets [102], wind-driven resuspension [103,104], and seasonal freshwater discharge [10,105]. The short-lived events may be restricted to only a few days each month and may occur sporadically, and as such, may be missed by observations from LEO satellites. The reasonable spatial similarity between monthly Max$_{\Delta TSS}$ and frequency of TSS above guideline thresholds suggests that diurnal changes are the main drivers of elevated TSS in the open coastal waters of the GBR.

Monthly frequency exceedance maps based on hourly Himawari-8 observations revealed that the open coastal and mid-shelf waters of the GBR lagoon present above thresholds TSS values (>2 mg L$^{-1}$) for wet and dry seasons. Although seasonal patterns were not visually conspicuous in the maps, the widespread exceedance of TSS during dry season months may be associated with the trade winds and resultant inshore wind waves, which resuspend fine sediments along the coastal waters of the central and south GBR [106–108]. In fact, significant hourly increases in turbidity associated with wind and wave conditions are well established on the inshore GBR [104]. In this context, wind stress is likely to be one of the major factors contributing to the dry season widespread exceedance of TSS at LJCO and, ultimately, to the long-term decline in inshore water clarity in the GBR [108,109].

Tides are usually considered a minor factor influencing the resuspension of bottom sediments and are more often associated with the periodical flushing of estuaries and bays into the GBR [106]. However, tides not only affect water level and circulation but also induce variations in the rates of sediment resuspension, mixing, and settling, contributing to changes in inshore water clarity and quality. Fabricius et al. [110] concluded that tides largely modulate the spatial complexity of long-term values of water quality parameters (i.e., photic depth) in the coastal GBR. In fact, the most persistent hotspots of TSS were confined to the outlets of the Hinchinbrook tidal channel, including at the LJCO. Large tidal ranges with associated strong tidal streams can resuspend fine sediments if they are available, with fines often remaining in suspension for many days until reduced hydrodynamic activity allows resettlement [111]. Strong relationships between TSS hotspots and tidal resuspension are thus likely, particularly where wet season discharge delivers fine, easily entrained sediment fractions.

Importantly, Fabricius et al. [110] points out that short-lived discharge events may have long-lasting impacts and reduce photic depth (water clarity) for many months following floods. In their work, Fabricius and colleagues recorded the shallowest photic depth in May, well after the peak of wet season floods. Their observations accord well with our results which indicate that April and May were the months in which TSS values were most frequently above guideline thresholds in the open coastal and mid-shelf waters near LJCO. Therefore, wet season freshwater discharge may be linked to the increased spatial extent of exceedances during the early dry season months (April and May), while seasonal wind patterns may be linked to the exceedances in the mid and late dry season months (June to September). Finally, the tidal ranges may be associated with semi-permanent features of high diurnal variability adjacent to the coastal areas. Further investigation and correlation with continuous weather and oceanographic data are needed for an improved understanding of the diurnal patterns of TSS in the GBR.

The exceedance maps were produced from 60 to ~170 valid observations, which is 2 to 8 times the number of images available from LEO satellites [38,112]. Although a low number of valid observations are expected during wet season months because of intensified cloud cover and sun glint, the number of Himawari-8 observations still significantly surpassed those available from LEO ocean colour sensors, which may deliver valid or not imagery only once-a-day.

Finally, our work demonstrates persistently exceedance (>80%) of guideline thresholds established for the open coastal and mid-shelf areas of the GBR. Concerns have been previously expressed by the Reef Plan Independent Science Panel (Reef Plan ISP) about the spatial and temporal insensitivities of the metric associated with annually averaging data over large areas. This is not surprising, considering daily ocean colour observations may introduce significant bias when calculating long-term trends [96] and may not reflect the actual coastal dynamics of the GBR. The water quality metric currently used for the Reef Report Card is derived from a biogeochemical model that assimilates daily atmospherically corrected surface reflectance from Sentinel-3A [52,60]. In this context, the water quality products derived at diurnal scales from Himawari-8 offer an opportunity to: (a) reassess the current methodology employed for tracking and mapping flood plumes [38]; (b) calculate the frequency that the GBR's ecosystems are exposed to freshwater discharge and pollutants [48,71,112]; and (c) increase the data available for validation and assimilation into biogeochemical and ocean colour models currently being employed for the GBR [47,61,62,113,114].

## 5. Conclusions

This work assessed the ability to quantify diurnal TSS variability from Himawari-8 AHI in the GBR. In general, Himawari-8 TSS products compared remarkably well to concurrent measurements of in situ TSS derived from particulate backscattering measurements at the LJCO. Significant hourly fluctuations of TSS (~7 mg L$^{-1}$) were shown at this site, insufficiently captured by once-per-day in situ measurements or by LEO ocean colour

observations. The diurnal fluctuations of TSS were primarily associated with the local tides and with freshwater discharge from the Herbert River. Moreover, Himawari-8 AHI hourly observations allowed the tracking and mapping of the Herbert River plume, revealing intricate coastal dynamics driving plume extent and direction. Additionally, the plume arrival was captured by a webcam installed at the LJCO, providing an integrated assessment of plume dynamics substantially backed by Himawari-8 AHI spatial observations. Hourly Himawari-8 AHI TSS products allowed the computation of monthly exceedance maps with quantitative information, minimal data gaps, and reliable accuracy for informed and data-driven monitoring. The frequency exceedance maps are easy to interpret and can be produced for the entire extent of the GBR and may assist in the continuous and systematic managing of the GBR marine park.

Himawari-8's high temporal resolution of 10 min is a striking advantage over the once/twice-a-day LEO satellite observations and current in situ monitoring capabilities. In addition, the ANN inversion algorithm is extremely fast (~minutes), allowing near-real-time operational retrievals of water quality parameters for the entire GBR area from Himawari-8 AHI 10 min data. Although this analysis was limited to a small subset of the GBR (for brevity and because of consistently available in situ data from LJCO), the results confirm the ability of Himawari-8 AHI and its identical successor on Himawari-9 (operational since December 2022) for improved monitoring of the coastal GBR. The outputs presented here provide high-frequent and accurate quantitative information that can be complementary to the methods currently implemented by the Marine Monitoring Program [5,71] to assist Marine Park management in monitoring GBR water quality. We will consider developing per-pixel uncertainties for a future update of the TSS algorithm, e.g., as proposed for neural networks by Schroeder et al. [52], to enable propagation and aggregation of uncertainties for possible inclusion into the guideline reporting metric of the GBR Marine Monitoring Program.

**Author Contributions:** Conceptualization, L.P.-V. and T.S.; methodology, L.P.-V. and T.S.; software, L.P.-V., T.S. and Y.Q.; validation, L.P.-V.; formal analysis, L.P.-V.; data curation, L.P.-V., T.S. and Y.Q.; writing—original draft preparation, L.P.-V.; writing—review and editing, T.S., M.J.D., S.S. and Y.Q.; supervision, T.S., M.J.D. and S.S.; funding acquisition, L.P.-V. All authors have read and agreed to the published version of the manuscript.

**Funding:** This research was funded by the National Council for Scientific and Technological Development (CNPq) Foundation of the Brazilian Federal Government through the Sciences without Borders Program, grant number 206339/2014-3.

**Data Availability Statement:** The data presented in this study are available on request from the corresponding author.

**Acknowledgments:** We acknowledge Juergen Fischer and Michael Schaale (Freie Universität Berlin) for providing access to the MOMO radiative transfer code and for the inverse modelling tool. We acknowledge Erin Kenna (CSIRO Environment) for his help with data processing. The Japan Meteorological Agency is acknowledged for the operation of Himawari-8 and data distribution through the Australian Bureau of Meteorology. The Australian Bureau of Meteorology is acknowledged for providing tidal prediction data. Data was sourced from Australia's Integrated Marine Observing System (IMOS)—IMOS is enabled by the National Collaborative Research Infrastructure Strategy (NCRIS). NCRIS (IMOS) and CSIRO are acknowledged for funding the Lucinda Jetty Coastal Observatory. This research was undertaken with the assistance of resources from the National Computational Infrastructure (NCI Australia), an NCRIS-enabled capability supported by the Australian Government.

**Conflicts of Interest:** The authors declare no conflict of interest.

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
