# Peer review of "Meteorological Satellite Observations Reveal Diurnal Exceedance of Water Quality Guideline Thresholds in the Coastal Great Barrier Reef"

_remotesensing, doi:10.3390/rs15092335_

Round 1

Reviewer 1 Report

The paper describes an integrated monitoring method for the Great Barrier Reef (GBR) marine protected area based on Himawari-8 Advanced Imager (AHI) observations and TSS computed from bio-optical measurements at the Lucinda Jetty Coastal Observatory (LJCO), Australia.

I would recommend the following:

Introduction – Please consider a brief geographical description of study area – it would help the readers not familiar with it. Split the first paragraph right after “Thirty-five major rivers drain into the GBR lagoon, comprising the largest external source of ‘new’ nutrients to the system [8].” – continue with brief geographical description, possibly bring here Figure 1.

Figure 1 – Please consider replacing Figure 1a with a topographic map of the region, including rivers’ network, as described in text – keep the red rectangle indicating location of study area sensu strictum. Keep Figures 1b and 1c – but consider a red arrow in both, indicating LJCO location. Mention both red rectangle and red arrow (?) in figures explanation.

For 2.1 The Lucinda Jetty Coastal Observatory - Please consider a more comprehensive description of the Herbert River plume.

Please consider a wider description of Sentinel-2 imaging usage in the context of your study.

Please improve the description of the satellite observations validation process (with in-situ measurements).

Author Response

Please see attachment, your comments were addressed in table ref "Reviewer 1". Thank you.

Reviewer 2 Report

The study sounds interesting. Please address the following comments to enhance some areas within the study.

Introduction: It is highly recommended to discuss advantages and disadvantages of your methodology of using Himawari-8 AHI

Figure 1 Caption: Replace (A and B) with (a and b) in line 117 (if you were referring to plots a and b)

Figure 2 looks busy, try to enhance for more clarity

Explain and justify why you selected 3*3 pixel in line 192-193

Table 1: Indicate how many years were used to estimate the annual and seasonal TSS in line 270

Line 342: Change Table 1 to Table 2

Figure captions: Check for consistent formats, look at Figures 9&10 compared to the figures before

Discussion: Lines 509 through 529 are unclear, try to enhance for more clarity

Author Response

Please see attachment. Your comments were addressed in table ref "Reviewer 2". Thank you.

Reviewer 3 Report

The authors submitted a well written and an interesting manuscript of the study dealing with water quality monitoring using geostationary satellite observations.

The methodology is well described and the conclusions are supported by the results. The manuscript could be accepted for publication after its revisions based on the input of all comments and suggestions provided by reviewers and Editors.

Author Response

Please see attachment. Your comments were addressed in table ref "Reviewer 3". Thank you.

Round 2

Reviewer 1 Report

No further remarks. From my point of view, the manuscript can be accepted in present form.